# High-Throughput Phenotyping for Agronomic Traits in Cassava Using Aerial Imaging

**DOI:** 10.3390/plants14010032

**Published:** 2024-12-25

**Authors:** José Henrique Bernardino Nascimento, Diego Fernando Marmolejo Cortes, Luciano Rogerio Braatz de Andrade, Rodrigo Bezerra de Araújo Gallis, Ricardo Luis Barbosa, Eder Jorge de Oliveira

**Affiliations:** 1Centro de Ciências Agrárias, Ambientais e Biológicas, Universidade Federal do Recôncavo da Bahia, Cruz das Almas 44380-000, Bahia, Brazil; henrique.bernardino.bio@gmail.com (J.H.B.N.); diegomarmol82@hotmail.com (D.F.M.C.); 2Embrapa Mandioca e Fruticultura, Nugene, Cruz das Almas 44380-000, Bahia, Brazil; lucianorogerio@gmail.com; 3Instituto de Geografia, Universidade Federal de Uberlândia, Av. João Naves de Ávila, 2121—Bairro Santa Mônica, Uberlândia 38408-902, Minas Gerais, Brazil; rodrigogallis@gmail.com (R.B.d.A.G.); rluisbarbosa@ufu.br (R.L.B.)

**Keywords:** *Manihot esculenta* Crantz, UAVs, prediction, vegetation indices

## Abstract

Large-scale phenotyping using unmanned aerial vehicles (UAVs) has been considered an important tool for plant selection. This study aimed to estimate the correlations between agronomic data and vegetation indices (VIs) obtained at different flight heights and to select prediction models to evaluate the potential use of aerial imaging in cassava breeding programs. Various VIs were obtained and analyzed using mixed models to derive the best linear unbiased predictors, heritability parameters, and correlations with various agronomic traits. The VIs were also used to build prediction models for agronomic traits. Aerial imaging showed high potential for estimating plant height, regardless of flight height (*r* = 0.99), although lower-altitude flights (20 m) resulted in less biased estimates of this trait. Multispectral sensors showed higher correlations compared to RGB, especially for vigor, shoot yield, and fresh root yield (−0.40 ≤ *r* ≤ 0.50). The heritability of VIs at different flight heights ranged from moderate to high (0.51 ≤ $H_{Cullis}^{2}$ ≤ 0.94), regardless of the sensor used. The best prediction models were observed for the traits of plant vigor and dry matter content, using the Generalized Linear Model with Stepwise Feature Selection (GLMSS) and the K-Nearest Neighbor (KNN) model. The predictive ability for dry matter content increased with flight height for the GLMSS model ($R^{2}$ = 0.26 at 20 m and $R^{2}$ = 0.44 at 60 m), while plant vigor ranged from $R^{2}$ = 0.50 at 20 m to $R^{2}$ = 0.47 at 40 m in the KNN model. Our results indicate the practical potential of implementing high-throughput phenotyping via aerial imaging for rapid and efficient selection in breeding programs.

## 1. Introduction

Cassava (*Manihot esculenta* Crantz) is a species known for its tuberous roots, which are of significant global importance, ranking as the fourth most important crop after wheat, rice, and corn [1]. It is a major source of starch for both human and animal consumption, as well as various industrial applications [1,2], and provides a key source of calories in developing countries, particularly in Africa and Asia [3]. Nigeria leads the world in cassava root production, followed by the Democratic Republic of Congo, Thailand, Ghana, and Brazil, with production figures of 60.00, 41.01, 28.99, 21.81, and 18.2 million tons, respectively [4].

The development of new cassava cultivars is essential for improving root yield, quality, and disease resistance [5]. However, cultivar selection in breeding programs is often hindered by conventional phenotypic evaluations, which are typically costly and have low throughput. This limitation makes it difficult to assess a large number of individuals, potentially reducing experimental precision and hindering the selection of superior genotypes that could maximize genetic gain. Therefore, adopting new approaches to overcome these challenges could improve data collection, enable larger breeding populations, and optimize the use of both financial and human resources [6].

Large-scale phenotyping using unmanned aerial vehicles (UAVs) is increasingly recognized as a key tool for improving genetic gains in breeding programs. UAVs provide valuable data for plant selection across various species [7,8]. They are effective in monitoring key growth parameters, such as biomass [9], leaf area index [10,11], chlorophyll content [12], vigor, and yield [13], all of which can be estimated using vegetation indices [14].

Vegetation indices are simple yet effective mathematical combinations that enable both quantitative and qualitative assessments of vegetation characteristics, such as ground cover, vigor, and growth rate [15]. These indices primarily rely on reflectance values from three wavelength ranges (visible, red edge, and near-infrared). These indices help reduce the volume of data to be analyzed, facilitating the estimation of structural and physiological biophysical variables of vegetation [16]. By combining reflectance values, vegetation indices link data to the physiological traits of plants [17].

Among the many vegetation indices, the Normalized Difference Vegetation Index (NDVI) is the most widely used [18]. It uses spectral reflectance from the red and near-infrared bands and is widely used due to its high sensitivity to changes in vegetation vigor [19]. NDVI is also useful for detecting and mapping the spatial distribution and temporal changes of vegetation [20]. Other indices, such as the Normalized Green–Red Difference Index (NGRDI), are valuable for estimating vegetation fraction, green biomass, leaf chlorophyll content, and plant phenology [21]. The Chlorophyll Vegetation Index (GLI), which differentiates live plants from dead ones and exposed soil, is effective in detecting leaf chlorophyll variations and assessing vegetation degradation [22].

In a recent study on cassava, Selvaraj et al. [23] found promising results in predicting fresh root yield using vegetation indices like the Green Normalized Difference Vegetation Index (GNDVI) and the Normalized Difference Red Edge Index (NDRE). Maresma et al. [22] tested several vegetation indices and discovered that the Wide Dynamic Range Vegetation Index (WDRVI) was the most accurate for predicting grain yield in corn (*Zea mays* L.). In wheat, Kyratzis et al. [24] analyzed vegetation indices across twenty varieties under water deficit conditions, reporting that GNDVI was the most effective predictor of grain yield when measured during the early reproductive stages.

Prediction models are crucial for identifying agronomically important traits and prioritizing those to be collected in the field, thereby reducing phenotyping costs in breeding programs [25]. In cassava cultivation, prediction models should focus on indices that reflect the genotypes’ ability for above-ground biomass production and, most importantly, root yield.

Several studies have shown that traits such as above-ground biomass, root diameter, and branching density can predict water and nutrient efficiency, making them valuable for selecting cassava genotypes during early growth phase [26]. However, implementing prediction models in agriculture presents challenges, including the lack of extensive databases, the high cost of sensors and technology, and the need for specialized training to develop and maintain these systems [27]. Despite these obstacles, as precision farming practices become more widespread and more agricultural data are collected and stored, the benefits of using prediction models will become more apparent. While prediction models for key traits are still in the early stages, initial results are promising [28].

One critical factor that impacts predictions is flight height, which directly affects the spatial resolution of the images [29]. Mesas-Carrascosa et al. [30] observed that the more images collected, the better the spatial resolution; in other words, spatial resolution is directly related to flight height and can be optimized for greater image detail.

Despite the growing use of UAVs in phenotyping, studies on agronomic traits and vegetation indices in cassava at various flight altitudes remain limited. For example, Selvaraj et al. [23] used flight altitudes between 30 and 40 m to assess four cassava genotypes, but they did not provide comparative results for these altitudes. Similarly, Rattanasopa et al. [31] used a flight altitude of 73 m to capture imagery of the “Kasetsart 50” cultivar. This study evaluates four distinct flight altitudes in genotypes from Embrapa’s cassava breeding program, providing critical insights for high-throughput phenotyping and emphasizing the importance of selecting optimal flight altitudes to improve data accuracy and quality in UAV-based phenotyping.

Given the increasing demand for efficient high-throughput phenotyping methods to enhance selection processes in cassava breeding, this study aims to (I) assess the correlation between agronomic data and vegetation indices obtained at various flight altitudes, including the accuracy of plant height measurements from digital elevation models; (II) identify the optimal flight altitude for evaluating cassava genotypes in terms of root yield and quality; and (III) develop predictive models for agronomic traits based on vegetation indices derived from aerial imagery at different flight altitudes.

## 2. Results

### 2.1. Heritability and Correlation Between Agronomic Traits and Vegetation Indices

The correlations and Bland–Altman agreement between plant heights measured conventionally and those derived from aerial images at four different flight heights are shown in Figure 1. The results reveal very high correlations between the two measurement methods (*r* = 0.99). The 95% limits of agreement captured 95% of the differences between the two methods, indicating strong concordance across all flight heights. However, measurement quality varied across the four flight heights. As flight height increased, a greater bias was observed in the estimated plant heights compared to ground measurements. Specifically, the difference was approximately 0.25 m at a flight height of 20 m and around 0.83 m at 60 m. This suggests that higher flight altitudes tend to overestimate plant height relative to ground-based measurements.

Flight altitude is directly related to mission duration and image processing time (Table 1). At a flight height of 60 m, fewer images were captured (50), and the flight duration was shorter (4 min) compared to 20 m, where nearly five times as many images were taken (222) and the mission duration was more than double (10 min). Shorter flights consume less battery, allowing for an increased number of missions and, consequently, more trials to be evaluated. However, in this study, a greater bias was observed in estimating plant heights at higher flight heights, which should be taken into account during field evaluations.

Moderate correlations (−0.40 ≤ *r* ≤ 0.50) were observed between vegetation indices from multispectral sensors and agronomic data for cassava during the 2022 crop season at a flight height of 60 m (Figure 2). Specifically, the vegetation indices DVI (*r* = 0.50), NDVI (*r* = 0.40), GNDVI (*r* = 0.40), RVI (*r* = 0.40), and CIG (*r* = 0.40) at 60 m showed moderate correlations with plant vigor. Similarly, the SCI and SI indices were moderately correlated with above-ground biomass yield (*r* = 0.40), while the NGRDI and VARI indices showed a similar correlation with fresh root yield (r = 0.40). In contrast, the indices BGI (*r* = −0.40 to −0.50), GLI (*r* = 0.40), and PSRI (*r* = −0.40 to −0.50) were associated with plant vigor at flight heights of 40 m and 60 m, respectively.

The correlation estimates between vegetation indices obtained from images captured by the RGB camera showed moderate correlations for traits such as plant vigor, leaf retention, dry matter content, and fresh root yield (Figure 3). Plant vigor exhibited a moderate correlation with the vegetation indices BI (*r* = −0.40), BGI (*r* = −0.40), and PSRI (*r* = −0.51). Similarly, leaf retention was correlated with the indices BI (*r* = −0.40) and CIRE (r = 0.50), while dry matter content (*r* = −0.40) and fresh root yield (*r* = 0.40) displayed moderate correlations with the HUE index. These correlations were observed in images captured at an altitude of 60 m. Additionally, the HUE vegetation index showed a moderate correlation (*r* = −0.40) at an altitude of 30 m with both leaf retention and dry matter content.

The heritability estimates ($H_{Cullis}^{2}$) for the eighteen indices obtained from the multispectral camera at four different flight heights (20 m, 30 m, 40 m, and 60 m) ranged from moderate ($H_{Cullis}^{2}$ = 0.30 to 0.60) to high ($H_{Cullis}^{2}$ ≥ 0.60) (Figure 4). Among these vegetation indices, CIRE ($H_{Cullis}^{2}$ = 0.82) and HI ($H_{Cullis}^{2}$ = 0.92) showed the highest heritability values at 20 m and 60 m, respectively. Additionally, indices such as GNDVI, NDVI, VARI, CIG, GLI, and CIRE exhibited high heritability (0.60 ≤ $H_{Cullis}^{2}$ ≤ 0.82) when measured at 20 m. At other flight heights, high heritability values were observed for the indices SCI, SI, RVI, and DVI at 30 m (0.57 ≤ $H_{Cullis}^{2}$ ≤ 0.76); BGI ($H_{Cullis}^{2}$ = 0.61) and BI ($H_{Cullis}^{2}$ = 0.56) at 40 m; and HUE, NGRDI, PSRI, CVI, NDRE, and HI (0.59 ≤ $H_{Cullis}^{2}$ ≤ 0.92) at 60 m.

The heritability estimates for the indices extracted from the RGB camera were similar to those obtained from the multispectral camera, ranging from moderate to high (Figure 5). The PSRI ($H_{Cullis}^{2}$ = 0.94) and HUE ($H_{Cullis}^{2}$ = 0.97) indices exhibited particularly high heritability. Other indices with high heritability included NDVI and GNDVI ($H_{Cullis}^{2}$ = 0.83 and $H_{Cullis}^{2}$ = 0.88, respectively) when measured at 20 m. At 30 m, the NDRE, SI, SCI, HI, and GLI indices displayed high heritability values (0.88 ≤ $H_{Cullis}^{2}$ ≤ 0.94). At 40 m, high heritability was observed only for the CIRE index ($H_{Cullis}^{2}$ = 0.83), while the remaining indices showed high heritability values (0.80 ≤ $H_{Cullis}^{2}$ ≤ 0.93) when measured at 60 m. Additionally, high heritability was observed for all agronomic traits ($H_{Cullis}^{2}$ > 0.90) (Table 2).

### 2.2. Prediction of Agronomic Traits Based on Vegetation Indices at Different Flight Height

The performance of prediction models for agronomic traits in cassava, using multispectral and RGB images captured by UAVs, was evaluated through cross-validation and variable selection based on the importance of the indices for prediction. The $R^{2}$ values for different flight heights indicated low prediction accuracy for the agronomic variables, ranging from 0.05 to 0.34 (Table 3).

Overall, the models performed better in predicting plant vigor and dry matter content, especially when using vegetation indices captured at a height of 20 m. The PLS and KNN models demonstrated higher predictive ability for plant vigor ($R^{2}$ = 0.30), while the PLS model yielded an $R^{2}$ = 0.34 for dry matter content. For leaf retention, the GLMSS model showed the best predictive performance ($R^{2}$ = 0.29), also utilizing vegetation indices captured at 20 m.

For leaf spot resistance, as well as fresh root yield and above-ground biomass yield, the best predictions were obtained using vegetation indices from flight heights of 60 m and 30 m. In particular, the SVM model achieved the highest prediction accuracy for fresh root yield and above-ground biomass yield ($R^{2}$ = 0.22), while the GLMSS model ($R^{2}$ = 0.16) produced comparable results at 60 m. For leaf spot resistance, the PLS model provided the best predictive performance ($R^{2}$ = 0.20) using images at 30 m.

The average relationship between observed and predicted values is presented in the Appendix A. The best fits were observed for plant vigor, with R^2^ = 0.50 using the KNN model and images collected at 20 m, and R^2^ = 0.47 for images taken at 40 m. For dry matter content, predictive accuracy improved, with R^2^ = 0.26 at 20 m and R^2^ = 0.44 at 60 m. While the models showed varying degrees of fit, most regressions were statistically significant (*p* ≤ 0.01), indicating a clear relationship between observed and predicted values.

## 3. Discussion

### 3.1. Efficiency of Measuring Cassava Plant Height Using Digital Elevation Models

This study showed a high correlation (*r* = 0.99) between plant height estimates from the digital elevation model based on UAV images. Similar results were found in cassava studies, with a strong correlation (*r* = 0.83) between UAV measurements at 30 m and 40 m and ground-based measurements [23]. While both point cloud and manual methods were consistent, lower flight height showed less bias compared to higher ones. From a practical standpoint, lower flight height results in more images being collected, which enhances the reliability of the results [32]. Based on our findings regarding cassava plant height, we recommend conducting UAV flights at an altitude of 20 m for the most accurate measurement of plant height.

### 3.2. Correlation Between Vegetation Indices and Agronomic Data in Cassava

The use of UAVs in high-throughput phenotyping has gained significant traction, particularly in breeding programs where large numbers of individuals must be assessed quickly for multiple traits. Despite the significant potential of UAVs, the correlation between vegetation indices and agronomic traits ranged from low to moderate (−0.1 ≤ *r* ≤ 0.5) at different flight heights. Generally, vegetation indices derived from multispectral sensors showed higher correlation magnitudes compared to those derived from RGB sensors, although the correlation remained low, especially at lower flight heights (20 m and 30 m).

Multispectral images are particularly advantageous for precision agriculture due to their ability to capture additional spectral information compared to RGB images, which typically offer lower resolution and color quality [32]. For example, in rice, Zhou et al. [14] explored the relationship between leaf area index and vegetation indices at different phenological stages and found that several indices derived from multispectral images had higher correlations (ranging from 0.63 to 0.79) compared to those from RGB images (ranging from 0.36 to 0.38).

In this study, the highest correlations (ranging from −0.4 to 0.5) were observed for traits such as plant vigor, above-ground biomass yield, fresh root yield, and dry matter content, particularly with indices derived from multispectral images at flight altitudes of 40 m and 60 m. The vegetation indices BGI, CIRE, DNVI, GLI, GNDVI, NDVI, NGRDI, PSRI, SCI, SI, and VARI provided the strongest correlations for these traits.

Phenotyping methods differ across plant species but generally focus on traits related to plant vigor, growth, and productivity, such as grains, fruits, and roots. For example, studies have shown moderate correlations between the GNDVI index and vegetation cover (0.64 ≤ *r* ≤ 0.66) and leaf area (0.40 ≤ *r* ≤ 0.59) in trials of common beans (*Phaseolus vulgaris*) grown under various irrigation treatments [33]. Other research indicates correlations of the NDVI (0.36 ≤ *r* ≤ 0.53) and GNDVI (*r* = 0.42 ≤ *r* ≤ 0.56) with grain yield in durum wheat (*Triticum turgidum* subsp. *durum*) [24]. In cassava, indices like NDRE, NDVI, and GNDVI have also demonstrated strong correlations with traits such as above-ground yield, vegetation cover, and fresh root yield [23].

Vegetation indices have also shown strong correlations with agronomic traits in other crops, depending on flight height. For instance, a study by Avtar et al. [34] evaluated NDVI and NDRE indices at different flight heights (20 m, 60 m, and 80 m). They found that flights at 60 m correlated well with traits such as canopy diameter and plant height of oil palm (*Elaeis guineensis*), while indices from images captured at 60 m and 80 m showed a high correlation with plant vigor.

Rattanasopa et al. [31] used UAVs to estimate agronomic traits in cassava over three different growth stages (5, 6, and 7 months after planting) using multispectral images. Their results revealed a high correlation (*r* = 0.95, 0.91, and 0.96) between the NDVI, RVI, and CIRE vegetation indices and agronomic traits such as leaf area, canopy height, and total fresh weight. While some authors argue that these indices are essential for assessing or classifying soil cover, detecting climate changes, and monitoring soil deficits [31,35], our study found significant correlations between the NDVI, RVI, and CIRE indices and the traits such as vigor and leaf retention in cassava trials, although the correlation magnitudes were generally low.

### 3.3. Influence of Flight Height on Vegetation Indices

Different flight heights can affect the spatial resolution of images and, consequently the quality of orthomosaics. Lower flight heights reduce the mapped area, leading to a higher number of images and longer data processing times for orthomosaic generation [36]. Therefore, optimizing flight height is crucial, especially in areas where environmental factors, such as sunlight glare and persistent cloud cover may hinder image quality.

Our results indicate that vegetation index values can vary with flight height, regardless of the sensor type used. Some studies have reported that increased flight height can impair the accuracy of information derived from images due to loss of detail [37], while lower flight heights (between 15 and 30 m) tend to provide sharper and more detailed images [38]. For example, Mesas-Carrascosa et al. [30] found no significant differences in vegetation index values, such as NDVI, when comparing images captured at higher flight heights.

Overall, our study showed greater variation in vegetation indices between different flight heights using RGB sensors, compared to indices obtained from multispectral images. This may be because cassava vegetation appears relatively homogeneous in RGB images, and the spatial variation from different flight heights can affect reflectance properties and image quality in the visible spectrum. In contrast, multispectral images undergo radiometric calibration before orthomosaic processing [39,40], resulting in less variation in vegetation indices derived from these images. Factors such as solar angle, time of day, bidirectional reflectance, and crop type can all influence vegetation index values. However, research on these factors remains limited, and more comprehensive studies are needed to fully understand the observed results.

The choice of flight height is typically determined by the study’s objectives and the target traits for inference. Current research is investigating the impact of flight height on vegetation index calculations and their correlation with traits in various crops. Some studies, such as those by Perroy et al. [41] and Quirós and Khot [42], found that higher flight height resulted in lower spatial resolution and poor canopy detection in *Miconia calvescens* and apple trees (*Malus domestica* Borkh), respectively. In contrast, our study observed increased vegetation index values for cassava when analyzing images from higher flight heights. Overall, a flight height of 60 m showed stronger correlations with most agronomic traits evaluated, such as dry matter content, fresh root yield, above-ground biomass yield, and plant vigor, indicating its high potential for future aerial imaging applications.

### 3.4. Heritability of Vegetation Indices and Agronomic Traits

Regardless of the sensor type (RGB or multispectral), the broad-sense heritability estimates for vegetation indices obtained at different flight heights ranged from moderate to high (0.51 ≤ $H_{Cullis}^{2}$ ≤ 0.94). The PSRI index ($H_{Cullis}^{2}$ = 0.94) and HUE ($H_{Cullis}^{2}$ = 0.97), derived from RGB sensors, exhibited the highest heritability estimates when measured at 30 m. Meanwhile, the multispectral indices CIRE ($H_{Cullis}^{2}$ = 0.82) and HI ($H_{Cullis}^{2}$ ≤ 0.92) showed the highest heritability values for images captured at 20 m and 60 m flight height, respectively. Silva et al. [43] reported high average heritability values (0.87) when using UAV-based phenotyping for wheat genotypes at crop maturation using multispectral indices.

Although high heritability values suggest that vegetation indices can be useful for the indirect selection of desirable attributes, such as yield traits in breeding programs, some studies report limited success in achieving high heritability for these indices. For example, Tao et al. [44] studied variations in growth and vegetation indices of slash pine (*Pinus elliottii*) at two locations, finding that heritability estimates for indices like the Landsat Soil Adjusted Vegetation Index (SAVI), GNDVI, and NDVI were very low (~0.11 at the first location and 0.23–0.27 at the second location) using RGB and multispectral cameras. This variability in heritability suggests that for cassava breeding, it is important to assess which vegetation indices are most stable and reliable across environments.

In elephant grass (*Cenchrus purpureus* (Schumach.) Morrone), heritability for vegetation indices and total dry biomass (TDB) was assessed using both basic linear mixed models and spatial linear models. Heritability values ranged from 0.22 (for MSAVI: Modified Soil-Adjusted Vegetation Index) to 0.55 (for GLI: Green Leaf Index). Generally, broad-sense heritability for individual indices was higher than for TDB, suggesting that certain vegetation indices could be used as secondary traits to support the indirect selection of superior genotypes. The NDRE index was particularly effective for indirect selection of TDB, with heritability 2.7 times greater than the trait itself, though genetic gains from indirect selection were lower than from direct selection. The efficiency of indirect selection depends on the heritability of the traits and their correlations [45].

Heritability is a population characteristic, not an individual one [46], so estimating heritability for vegetation indices at different stages of a breeding program is essential to define effective selection strategies. In uniform yield trials, where replications, locations, and years are considered, more accurate heritability estimates are obtained compared to preliminary trials, which rely on averages of replications [47]. Understanding these correlations is crucial for applying selection based on vegetation indices estimated through high-throughput phenotyping via UAVs, allowing breeding programs to fully exploit the potential of this approach for breeding. By selecting indices with high heritability that are correlated with desirable traits, cassava breeders can use high-throughput phenotyping techniques to identify superior genotypes more efficiently. This can significantly improve the speed of breeding cycles, reduce costs, and help develop cassava varieties that are higher-yielding, more disease-resistant, and better adapted to changing environmental conditions.

Similar to the vegetation indices, agronomic traits showed high heritability estimates (0.93 ≤ $H_{Cullis}^{2}$ ≤ 0.99), likely due to the analysis of trials in the later stages of agronomic validation (UYT), where plots contained more plants, leading to greater uniformity across different replications. Similar findings were reported by Sampaio Filho et al. [48], who presented $H_{Cullis}^{2}$ values of 0.94, 0.94, and 0.93 for above-ground biomass yield, fresh root yield, and dry matter content, respectively. Additionally, Conceição et al. [49] reported $H_{Cullis}^{2}$ values of 0.84 for leaf retention and 0.72 for plant vigor.

### 3.5. Performance of Vegetation Index-Based Models for Predicting Agronomic Traits in Cassava

Prediction models based on vegetation indices are valuable tools for identifying relationships between remote sensing data and agronomic parameters, such as yield, leaf area index, and biomass [50,51]. In this study, machine learning models demonstrated low to moderate predictive abilities for various cassava traits (0.01 ≤ R^2^ ≤ 0.50).

The highest predictive performance was observed for plant vigor and dry matter content using the GLMSS and KNN models. For plant vigor, the KNN model achieved $R^{2}$ values of 0.50 and 0.47 at flight altitudes of 20 m and 40 m, respectively. For dry matter content, the GLMSS model showed improved predictive ability with $R^{2}$ = 0.26 at 20 m and $R^{2}$ = 0.44 at 60 m. Conversely, the PLS model consistently yielded the lowest $R^{2}$ values for all agronomic traits. These results contrast with previous studies, such as Wang et al. [50], who reported better performance of the PLS model for several variables. Similarly, while Li et al. [52] demonstrated the high predictive ability of the SVM model for wheat yield during the grain-filling stage ($R^{2}$ = 0.73 and RMSE = 0.87 t/ha) using hyperspectral imagery, our study found lower SVM performance for cassava traits (0.01 ≤ $R^{2}$ ≤ 0.26) using spectral images.

The predictive accuracy of these models is influenced by flight altitude, which affects imaging parameters like spatial resolution and field of view. Lower altitudes (e.g., 20 m) provide higher spatial resolution, crucial for capturing fine details such as leaf texture and spectral variations. Studies by Guo et al. [53] and Ye et al. [54] highlight the importance of high-resolution imagery for detecting biological symptoms like yellow rust in wheat or *Fusarium* wilt in bananas. Conversely, higher altitudes (e.g., 60 m) offer broader coverage but compromise spatial resolution, limiting the ability to detect fine-scale variability. At 20 m, the KNN model achieved its best performance for plant vigor (R^2^ = 0.50). This aligns with findings from Wang et al. [55], who noted that KNN performs well with high-resolution images for traits influenced by texture and color differences captured through vegetation indices. However, while KNN performed well for above-ground traits, its predictive ability for below-ground traits, such as root yield, remained low. This reflects the inherent challenges of using aerial imaging to capture traits not directly observable through vegetation indices.

Similarly, PLS achieved moderate performance for plant vigor at 20 m (R^2^ = 0.30) showing better use of spectral variations, as demonstrated in studies by Lizuka et al. [56], where detailed spectral data enhanced predictions of vegetation cover fractions using regression models. As the flight altitudes increase to 30–40 m, the balance between spatial resolution and field of view begins to shift. At these altitudes, aggregation effects can reduce variability, improving correlations for certain traits, such as canopy cover, but at the expense of losing fine-scale details. For instance, the PLS model performed best at 30 m for predicting leaf spot resistance ($R^{2}$ = 0.20), effectively capturing spectral differences linked to leaf health. Similarly, the SVM model showed the highest predictive ability for fresh root yield and above-ground biomass yield at 30 m ($R^{2}$ = 0.22), benefiting from the moderate spatial detail suitable for biomass-related traits.

At the highest flight altitude of 60 m, reduced spatial resolution negatively impacted models that rely on detailed textures, such as PLS, whose performance for predicting dry matter content declined to $R^{2}$ = 0.10. However, models like GLMSS showed improved performance results at this altitude for traits dependent on broader spatial patterns, achieving an $R^{2}$ = 0.16 for fresh root yield.

The development of rapid phenotyping methods has become a priority in cassava breeding programs, aiming to shorten selection cycles, evaluate more genotypes, and reduce costs associated with measuring challenging traits. While significant progress has been made, few studies have focused on validating predictive models for root yield and dry matter content using image analysis in cassava. In this context, the predictive ability of $R^{2}$ = 0.44 for dry matter content achieved with the GLMSS model at a flight height of 60 m represents a promising starting point for early plant selection in breeding programs. Although this level of accuracy is moderate, it offers practical insights into traits that are otherwise difficult to measure directly.

These results highlight that while predictive accuracies remain moderate, aerial imaging shows considerable potential as a complementary tool to advanced methodologies like genomic selection, which has achieved similar levels of accuracy in certain cases [57,58]. Integrating aerial imaging with genomic selection could significantly enhance cassava breeding efforts by offering a cost-effective and scalable approach to evaluating complex traits. For instance, aerial imaging facilitates rapid assessments that align well with breeding objectives, particularly for traits such as root yield and dry matter content. However, the limited success in predicting root yield underscores the need for further methodological advancements.

### 3.6. Applications of Aerial Imaging in Cassava Breeding

The large volume of data generated by UAVs, particularly when equipped with sensors capturing extensive datasets, presents challenges in processing and delivering results quickly [59]. Many recent studies focus on validating spectral vegetation indices in field-scale plant phenotyping, with a significant portion of measurements taken proximally [60], which ties into the aforementioned challenges. Nevertheless, UAV-based studies have made notable progress in overcoming current limitations and maximizing the potential of this technology in agriculture. Despite obstacles like adverse weather conditions, sensor calibration issues, and platform stability—factors directly affecting data reliability and accuracy [59,60]—UAVs are expected to become a valuable tool for supporting breeding programs and aiding farmers in obtaining reliable precision agriculture data for informed decision making.

Breeding programs are actively developing technologies and sensor algorithms to improve data capture accuracy and precision. These advancements aim to enhance the detection and quantification of plant growth, chlorophyll content, plant height, and vigor, along with the use of state-of-the-art analytical techniques. In the context of the cassava breeding program, UAV-based phenotyping for estimating plant height has already proven feasible. However, the prediction of agronomic traits through aerial imaging phenotyping across various trials in a breeding program can still be enhanced to more effectively select individuals in large populations.

Correlations between agronomic traits and vegetation indices derived from RGB and multispectral sensors varied from low to moderate at different flight heights, contributing to modest predictive accuracies for most of the agronomic traits evaluated. To improve the predictive models, incorporating spatial features could help standardize performance across different camera systems. Similar findings were reported by Herzig et al. [61], who compared RGB and multispectral camera systems for predicting barley grain yield. Due to costs and simpler image processing, they suggested that RGB cameras might be preferred over multispectral systems.

Several factors should be considered for effectively integrating aerial imaging into cassava breeding selection routines. One potential approach is to capture images throughout the entire crop cycle, which would allow for the observation of environmental variations that influence productive traits. This strategy could result in more robust vegetation indices for analysis and ensure the standardization of the study area, ultimately improving the quality of the images captured.

To effectively integrate aerial imaging into cassava breeding programs, it is essential to capture images throughout the crop cycle, allowing for the observation of environmental variations that influence productive traits. This strategy could lead to the development of more robust vegetation indices, standardization of study areas, and improvements in image quality. The combination of aerial imagery with ground-based sensors and proximal sensing technologies could further enhance prediction accuracy. Moreover, incorporating data from multiple sources—such as spectral indices and environmental variables collected at key phenological stages—has the potential to significantly improve model performance for complex traits like root yield. Future research should focus on developing robust models that leverage these integrated data streams to refine predictive capabilities.

Despite existing challenges, advancements in UAV technologies and methodologies offer substantial promise for cassava breeding. By enabling rapid and cost-effective phenotyping, UAVs can play a pivotal role in evaluating complex traits and accelerating the selection process in large breeding populations.

## 4. Conclusions

Aerial imaging has demonstrated high accuracy for estimating traits such as plant height (*r* = 0.99), irrespective of flight altitude. Moderate agreement between aerial imaging and ground measurements was most notable at a flight height of 20 m. However, the predictive capability for below-ground traits, including root yield and dry matter content, remains limited.

Models like GLMSS showed moderate predictive ability for dry matter content at 60 m ($R^{2}$ = 0.44), but prediction accuracy for root yield was consistently low across all models and flight heights. These findings highlight the challenges of using aerial imaging to estimate below-ground traits, emphasizing the need for further methodological advancements or the integration of complementary data sources.

A flight height of 60 m provides practical benefits, such as reduced battery consumption, shorter flight durations, and greater field coverage, making it a feasible choice for phenotypic assessments in cassava breeding programs despite limitations in trait prediction accuracy.

In conclusion, aerial imaging is a promising tool for field phenotyping, particularly for above-ground traits like plant vigor. However, its effectiveness for predicting below-ground traits is constrained. Continued research is essential to improve predictive models and integrate additional data sources, thereby enhancing the utility of aerial imaging for cassava breeding programs.

## 5. Materials and Methods

### 5.1. Plant Material and Experimental Design

The experiment was conducted in June 2021 at Fazenda Botelho (latitude 11°48′7.24″ S, longitude 38°22′27.53″ W, with an average altitude of 224 m) in the rural area of Inhambupe, Bahia, Brazil. This region has a hot, subhumid tropical climate, with an average annual temperature of 34 °C, a mean annual relative humidity of 82%, and an average annual rainfall of 1100 mm, concentrated between March and August, followed by a warmer period from September to February. A total of 36 cassava clones from the uniform yield trial (UYT) were evaluated, representing the final stage in the process of evaluation and selection. A list of clones and their genealogy is provided in Appendix A.

UYTs are a critical phase in cassava breeding programs, where promising clones are assessed through multiple repetitions across diverse environments and over several years. This approach allows for higher experimental precision, particularly for traits with low heritability. These trials are designed to maximize data accuracy and reliability, providing a robust foundation for the selection of superior, well-adapted clones. The experimental design was a randomized complete block (RCBD) with three replicates, with plots consisting of three rows of seven plants each.

Soil preparation included plowing followed by two harrowings, after which planting furrows approximately 15 cm deep were created. Planting was performed manually during the rainy season in May 2021, using 16 cm seed stem cuttings with a spacing of 0.90 m between rows and 0.80 m between plants. Crop management practices followed recommended guidelines for cassava cultivation [62].

### 5.2. Evaluation of Agronomic Traits

In addition to vegetation indices at different flight altitudes, the following agronomic traits were evaluated 12 months after planting: (I) plant height (m), measured manually on 15 plants per plot using a tape measure from ground level to the apical meristem (shoot tip); (II) fresh root yield (FRY), obtained by weighing roots produced per plot on a hydrostatic scale (t.ha^−1^); (III) shoot yield (SY), determined by weighing all above ground biomass (stems, leaves, and petioles) from plants in the plot (t.ha^−1^); (IV) leaf spot resistance (LSR), visually assessed using a severity scale: 0 = no symptoms, 1 = 25% of leaves affected in the lower third of the plant, 2 = >50% of leaves affected in the lower third of the plant, 3 = leaves affected in the middle and lower thirds parts, 4 = mild incidence distributed throughout the plant, 5 = moderate incidence distributed throughout the plant, along with yellowing and/or defoliation of the lower third, 6 = complete defoliation of the plant; (V) leaf retention (LR), scored on a scale representing leaf cover at the apical meristem: 1 = less than 5% leaf retention, 2 = 6–15% leaf retention, 3 = 16–30% leaf retention, 4 = 31–50% leaf retention, 5 = more than 50% leaf retention; (VI) plant vigor (PV), assessed on a scale where 1 = low vigor, 3 = intermediate vigor, and 5 = high vigor; (VII) dry matter content (DMC) in roots (%), determined by weighing 3–5 kg of roots per plot to obtain air and water weights, following the gravimetric method proposed by Kawano et al. [63].

### 5.3. Image Acquisition

The acquisition of RGB and multispectral images was carried out using the DJI Phantom 4 Pro V2 UAV (DJI, Shenzen, China). The RGB camera has a focal length of 8.8 mm, a 5.5-inch display, and a 1-inch CMOS sensor with a resolution of 20 megapixels. The Micasense multispectral camera model RedEdge-M (MicaSense Inc., Seattle, WA, USA) features a fixed focal length of 5.4 mm, with sensors that have a resolution of 1280 × 980 pixels and five different spectral bands (Appendix A).

The flights were conducted 12 months after planting, on the same day as the ground measurements of plant height. Image collection with the UAV took place under favorable weather conditions, with a uniform sky between the hours of 10 a.m. and 2 p.m., which corresponds to the period of maximum direct solar irradiation on the trial. The DJI Pilot 2 app (DJI, Shenzen, China) was used to control the UAV’s flight path and speed during phenotyping, ensuring a 90% overlap between images.

Images were captured at different flight heights (20, 30, 40, and 60 m) to estimate the optimal flight height for predicting phenotypic data. To enhance the accuracy of the location of the experimental plots prior to each flight, ground control points (GCPs) were established by positioning targets around and within the area. GCPs are essential for improving the georeferencing accuracy of UAV-based imagery. By establishing GCPs around and within the experimental plots, the spatial alignment of the captured images is corrected, ensuring that the data corresponds accurately to real-world coordinates. The Emlid Reach RS+ GNSS system (Emlid Ltd, Budapest, Hungary), which offers 2 cm accuracy using real-time kinematic (RTK) technology, was used to measure these points. This high precision is critical for reducing distortions caused by variations in flight height, such as those experienced when capturing images at 20, 30, 40, and 60 m. The 19 ground control points (GCPs) were evenly distributed across the plantation, including its edges and interior, to ensure comprehensive coverage throughout the experiment.

### 5.4. Orthomosaic Construction, Processing, and Radiometric Calibration

The images were processed using Agisoft Metashape software version 1.5.5 (Agisoft LLC, St. Petersburg, Russia), following the workflow for image processing. This included alignment, generation of sparse point clouds from photo alignment, preparation of dense clouds, creation of a mesh for 3D visualization, image sharpening, orthorectification, and export of the final orthomosaics from the field [64].

Radiometric calibration was performed to produce multispectral and hyperspectral images that provide information consistent with a known radiometric reference. The preprocessing of images from the MicaSense RedEdge sensor utilized the radiometric calibration conversion formula provided by MicaSense (https://support.micasense.com/hc/en-us/articles/115000351194-RedEdge-Camera-Radiometric-Calibration-Model (accessed on 5 January 2024), which converts digital number (DN) values into absolute spectral radiance values based on the formula: $L=V\left(x,y\right)\times \frac{a_{1}}{g}\times \frac{{p-p}_{BL}}{t_{e}+a_{2}y-a_{3}t_{e}y}$, where L is the spectral radiance; $V\left(x,y\right)$ is the polynomial function of the vignette at pixel $\left(x,y\right)$; $a_{1}$, $a_{2}$ and $a_{3}$ are the radiometric calibration coefficients; g is the sensor gain setting; p is the normalized DN value; $p_{BL}$ is the black level offset; and $t_{e}$ is the exposure time of the image. All parameters necessary for calculating spectral radiance were obtained from the photo metadata. Considering various climatic and lighting conditions, radiometric calibration produces more accurate and reliable data that enable time series analysis and comparison of results over time.

MicaSense provides albedo values for the calibration panel, along with a CSV file containing the reflectance values for the panel, which are essential for radiometric calibration. These values were used in Agisoft Metashape during the image calibration process to correct for any variations in reflectance caused by environmental factors such as lighting, sensor conditions, and atmospheric influences. The albedo values represent the inherent reflectivity of the calibration panel, which is a known reference, and they allow for the accurate conversion of digital number (DN) values from the sensor into absolute spectral radiance values. By incorporating these values into the calibration process, Agisoft Metashape adjusts the raw imagery to more reliably reflect the true spectral characteristics of the captured scene, ensuring consistency and accuracy in the final processed data. The multispectral images captured with the RedEdge camera are radiometrically calibrated using Agisoft Metashape software. This software utilizes data recorded by the irradiance sensor, stored in the EXIF metadata of each image, to convert digital numbers into radiance and subsequently into reflectance before generating the orthophoto.

### 5.5. Estimation of Plant Height from Aerial Images

Plant heights were measured through digital image processing using QGIS software (version 3.32.3) and the Terrain Profile plugin. Data processed in Agisoft Metashape (version 1.5.5) were exported to generate the digital elevation model (DEM) and to crop the orthophoto. The orthophoto was overlaid with 50% transparency onto the DEM for each experiment, allowing for a clear visual comparison. A profile line was then drawn for each plant identified in the RGB image and verified against the field map for accuracy. This line was positioned to intersect both the lowest point on the DEM, representing ground level, and the highest point on the plant, enabling height estimation.

### 5.6. Acquisition of Vegetation Indices

Vegetation indices are mathematical formulations derived from spectral data collected via remote sensing, primarily in the red (R) and near-infrared (NIR) bands. In this study, we obtained 18 commonly used vegetation indices that are valuable for assessing plant vigor, growth, and predictive analysis (Table 4). These indices were calculated using the FIELDimageR package [64] in R software [65], which facilitated soil and weed removal through image segmentation with the fieldMask function.

### 5.7. Data Analysis

Agronomic traits at various flight heights (20, 30, 40, and 60 m) were evaluated using mixed models, defined by the formula Y=Xβ+Zg+ε, where y is the vector of phenotypic observations, β is the vector of fixed effects for blocks plus the overall mean, g is the vector of genotype-adjusted means with random effects, and ε is the vector of residuals, also treated as random effects. The matrices X and Z connect the independent variables to the response variable y.

Model effects were estimated using the lme4 package [79] in R software [65], with variance components calculated via restricted maximum likelihood (REML), and the best linear unbiased predictors (BLUPs) obtained for the random genotype effects. The significance of the model effects was tested using deviance analysis based on the likelihood ratio test (LRT) with a $\chi^{2}$ distribution at a 5% significance level.

Broad-sense heritability was estimated following the formula proposed by Cullis et al. [80] as an alternative expression for unbalanced data and mixed models: $H_{Cullis}^{2}=1-\frac{{\bar{V}}_{\Delta }^{Blup}\;}{2*\sigma_{g}^{2}}$, where $\sigma_{g}^{2}$ is the genetic variance and ${\bar{V}}_{\Delta }^{Blup}$ is the mean standard error of the genotypic BLUPs.

Pearson correlations were calculated to assess relationships between vegetation indices and agronomic characteristics obtained via UAV phenotyping, with significance determined using the *t*-test with n-2 degrees of freedom, via corrplot package in R software [65]. To assess the agreement between ground-measured plant heights and point cloud heights at different flight heights (20, 30, 40, and 60 m), we used the Bland–Altman [81] method. This method analyzes the agreement between two different measurement techniques assessing the same variable in the same units, allowing for the determination of whether the differences in measurements from the two methods are acceptable and equivalent. The Bland–Altman plot depicted the differences between each pair of observations (ground plant height—point cloud height) on the vertical axis, while the average of the pairs of observations [(ground plant height + Point cloud height2)/2] was plotted on the horizontal axis. The 95% limits of agreement were established as ±1.96 times the standard deviation (SD) of the bias. The Bland–Altman plot was created using the dplyr [82] and ggpubr [83] packages in R.

Vegetation indices were utilized to assess the predictive ability of agronomic traits using four prediction models: Generalized Linear Model with Stepwise Feature Selection (GLMSS), Partial Least Squares (PLS), Support Vector Machine (SVM), and K-Nearest Neighbor (KNN). The models were cross-validated using a scheme with 5 repetitions and 6 folds per repetition. A total of 60% of the samples were allocated for training, with the remaining 40% used for validation. All predictive models were implemented in the caret package [84] in R version 4.3 [65].

Model performance was evaluated based on the root mean square error (RMSE) and the coefficient of determination ($R^{2}$) for each cross-validation fold. RMSE measures the average magnitude of the estimated errors; the closer its value is to 0, the better the quality of the estimates. The RMSE was computed using the formula: $RMSE=\sqrt{\frac{1}{n}\sum_{i=l}^{n}(y_{i}-{\hat{y}}_{i\;}{)}^{2}}$, where $y_{i\;}$ and ${\hat{y}}_{i\;}$ are the observed and predicted values, respectively, and n is the number of observations. The $R^{2}$ statistic represents the proportion of the total variation in the dependent variable explained by the variation in the independent variable, estimated as follows: $R^{2}$ = $\frac{\sum_{i=l}^{n}({\hat{y}}_{i\;}-\bar{y}{)}^{2}}{\sum_{i=l}^{n}(y_{i\;}-\bar{y}{)}^{2}}$, where $\sum_{i=l}^{n}({\hat{y}}_{i\;}-\bar{y}{)}^{2}$ and $\sum_{i=l}^{n}(y_{i\;}-\bar{y}{)}^{2}$ correspond to the explained and unexplained variance by the model, respectively, with $\bar{y}$ being the mean of $y_{i}$.

## Figures and Tables

**Figure 1 plants-14-00032-f001:**
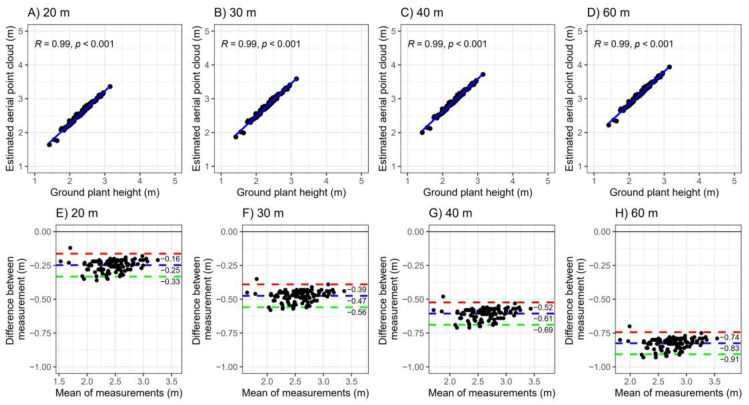
(**A**–**D**) Correlation graphs between ground plant heights and those obtained via point cloud from aerial images captured by the unmanned aerial vehicle (UAV) at different flight heights (20 m, 30 m, 40 m, and 60 m). (**E**–**H**) Bland–Altman agreement graphs comparing manually measured plant heights with those obtained from point clouds at different flight heights. The blue dashed line indicates the mean difference (bias) between the measurement methods, and the red (upper) and green (lower) lines represent the 95% limits of agreement (±0.96 × standard deviation).

**Figure 2 plants-14-00032-f002:**
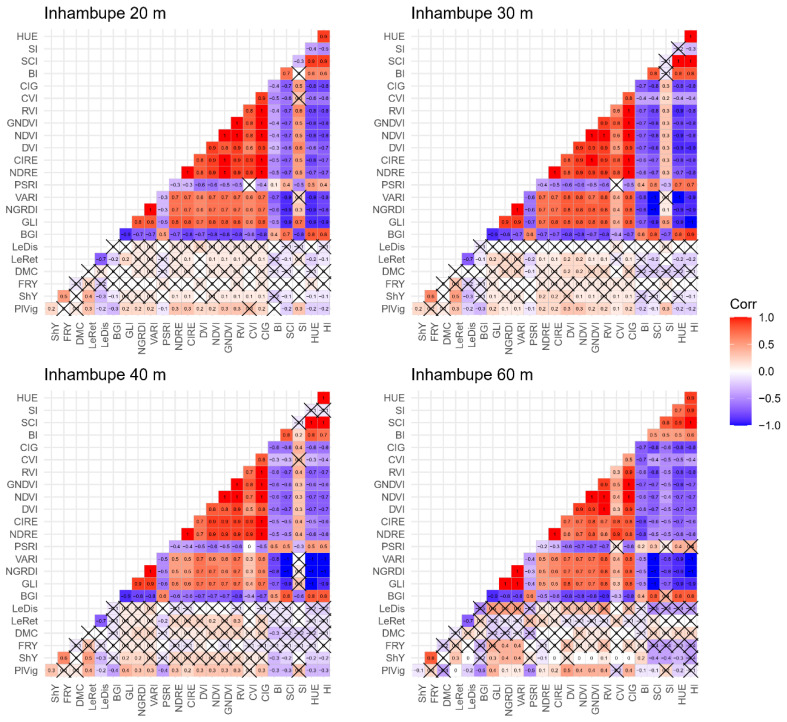
Correlations between vegetation indices obtained from aerial images captured by multispectral cameras on an unmanned aerial vehicle (UAV) at flight heights of 20 m, 30 m, 40 m, and 60 m, alongside agronomic characteristics. PlVig: plant vigor; ShY: above-ground biomass yield; FRY: fresh root yield; DMC: dry matter content in roots; LeRet: leaf retention; LeDis: leaf spot resistance; BI: Brightness Index; BGI: Blue Green Pigment Index; GLI: Green Leaf Index; HI: Primary Colors Hue Index; HUE: Hue Index; NGRDI: Normalized Green–Red Difference Index; SCI: Soil Color Index; SI: Spectral Slope Saturation Index; VARI: Visible Atmospherically Resistant Index; PSRI: Plant Senescence Reflectance Index; NDRE: Normalized Difference Red Edge Index; CIRE: Red-edge chlorophyll index; DVI: Difference Vegetation Index; NDVI: Normalized Difference Vegetation Index; GNDVI: Green normalized differences vegetation index; RVI: Ratio Vegetation Index; CVI: Chlorophyll Vegetation Index; CIG: Chlorophyll Index—green. The symbol × in the values indicates that the correlation was not significant according to the *t*-test at a 5% significance level.

**Figure 3 plants-14-00032-f003:**
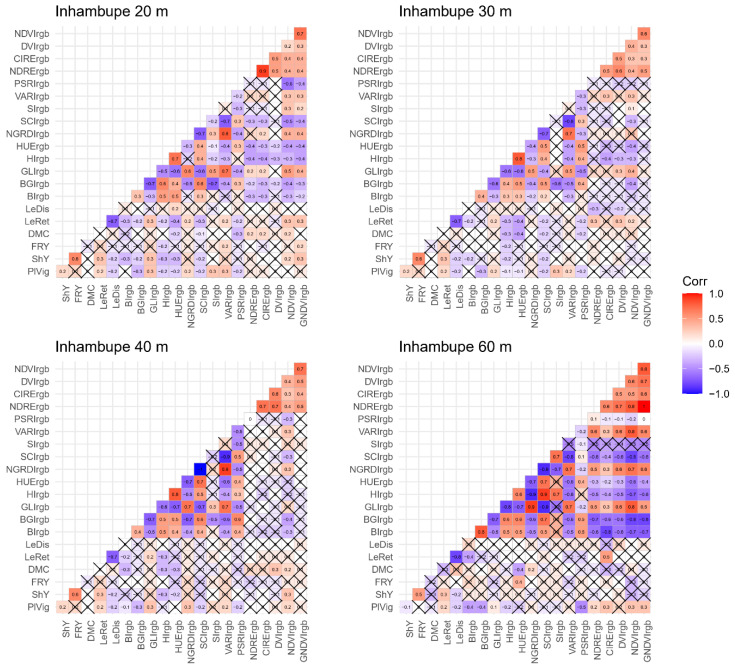
Correlations between vegetation indices obtained from aerial images captured by RGB (Red, Green, Blue) cameras mounted on an unmanned aerial vehicle (UAV) at flight heights of 20 m, 30 m, 40 m, and 60 m, alongside various agronomic traits. PlVig: plant vigor; ShY: above-ground biomass yield; FRY: fresh root yield; DMC: dry matter content in roots; LeRet: leaf retention; LeDis: leaf spot resistance. BI: Brightness Index (BI); BGI: Blue Green Pigment Index; CLI: Green Leaf Index (GLI); HI: Primary Colors Hue Index; HUE: Hue Index; NGRDI: Normalized Green–Red Difference Index; SCI: Soil Color Index; SI: Spectral Slope Saturation Index; VARI: Visible Atmospherically Resistant Index; PSRI: Plant Senescence Reflectance Index; NDRE: Normalized Difference Red Edge Index (NDRE); CIRE: Red-edge chlorophyll index; DVI: Difference Vegetation Index; NDVI: Normalized Difference Vegetation Index; GNDVI: Green normalized difference vegetation index. The symbol × indicates that the correlation was not significant according to the *t*-test at a 5% significance level.

**Figure 4 plants-14-00032-f004:**
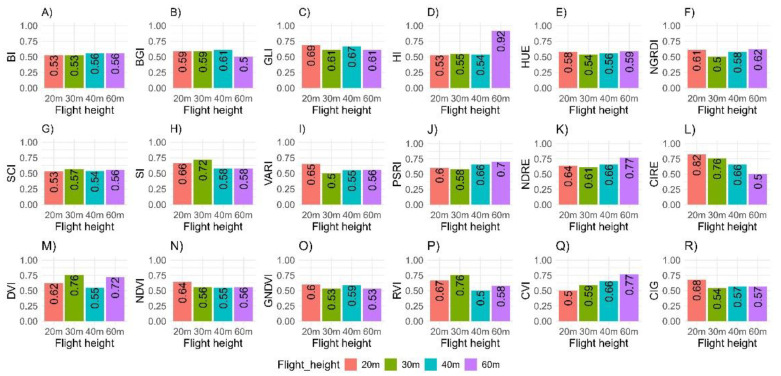
Estimates of broad-sense heritability ($H_{Cullis}^{2}$) for 18 vegetation indices obtained from aerial images captured by multispectral cameras using an unmanned aerial vehicle (UAV) at different flight heights (20 m, 30 m, 40 m, and 60 m). (**A**) Brightness Index (BI); (**B**) Blue Green Pigment Index (BGI); (**C**) Green Leaf Index (GLI); (**D**) Primary Colors Hue Index (HI); (**E**) Hue Index (HUE); (**F**) Normalized Green–Red Difference Index (NGRDI); (**G**) Soil Color Index (SCI); (**H**) Spectral Slope Saturation Index (SI); (**I**) Visible Atmospherically Resistant Index (VARI); (**J**) Plant Senescence Reflectance Index (PSRI); (**K**) Normalized Difference Red Edge Index (NDRE); (**L**) Red-edge chlorophyll index (CIRE); (**M**) Difference Vegetation Index (DVI); (**N**) Normalized Difference Vegetation Index (NDVI); (**O**) Green normalized differences vegetation index (GNDVI); (**P**) Ratio Vegetation Index (RVI); (**Q**) Chlorophyll Vegetation Index (CVI); (**R**) Chlorophyll Index—green (CIG).

**Figure 5 plants-14-00032-f005:**
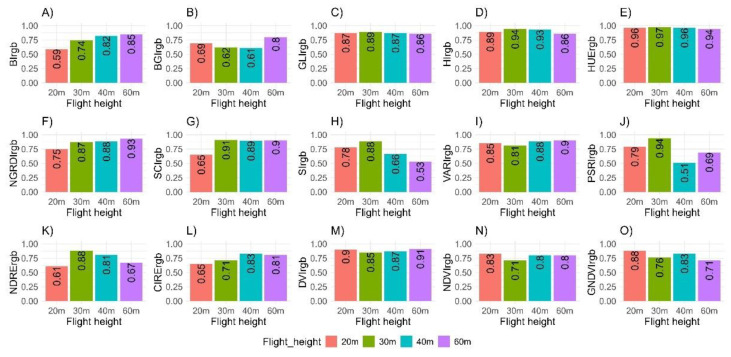
Estimates of broad-sense heritability ($H_{Cullis}^{2}$) for 15 vegetation indices obtained from aerial image analysis using RGB (Red, Green, Blue) cameras mounted on an unmanned aerial vehicle (UAV) at different flight heights (20 m, 30 m, 40 m, and 60 m). (**A**) Brightness Index (BI); (**B**) Blue Green Pigment Index (BGI); (**C**) Green Leaf Index (GLI), (**D**) Primary Colors Hue Index (HI); (**E**) Hue Index (HUE); (**F**) Normalized Green–Red Difference Index (NGRDI); (**G**) Soil Color Index (SCI); (**H**) Spectral Slope Saturation Index (SI); (**I**) Visible Atmospherically Resistant Index (VARI); (**J**) Plant Senescence Reflectance Index (PSRI); (**K**) Normalized Difference Red Edge Index (NDRE); (**L**) Red-edge chlorophyll index (CIRE); (**M**) Difference Vegetation Index (DVI); (**N**) Normalized Difference Vegetation Index (NDVI); (**O**) Green normalized difference vegetation index (GNDVI).

**Table 1 plants-14-00032-t001:** Summary of flight parameters and information on image collection and processing.

Flight Altitude (m)	Number of Images	Flight Duration	Flight Time (h/m/s)	Processing Time (RGB) h/m	Processing Time (Multispectral) h/m
20	222	10 min	10:17:46	02:52	02:12
30	129	7 min	10:00:10	02:12	01:44
40	69	5 min	09:46:56	01:32	01:29
60	50	4 min	09:29:29	01:22	01:15

h: hours; m: minutes; s: seconds.

**Table 2 plants-14-00032-t002:** Estimates of broad-sense heritability ($H_{Cullis}^{2}$) for the main agronomic traits in cassava.

$H_{Cullis}^{2}$					
PlVig	ShY	FRY	DMC	LeRet	LeDis
0.93	0.98	0.98	0.99	0.98	0.98

PlVig: plant vigor; ShY: above-ground biomass yield.; FRY: fresh root yield; DMC: dry matter content in roots; LeRet: leaf retention; LeDis: leaf spot resistance.

**Table 3 plants-14-00032-t003:** Accuracy and prediction error for certain agronomic traits in cassava using vegetation indices obtained at different flight heights from unmanned aerial vehicles.

Traits	Model ^1^	Flight Height (m)							
		20		30		40		60	
		RMSE	$R^{2}$	RMSE	$R^{2}$	RMSE	$R^{2}$	RMSE	$R^{2}$
Plant vigor	GLMSS	0.84	0.19	0.94	0.13	0.83	0.20	0.97	0.13
	KNN	0.80	0.30	0.91	0.13	0.80	0.26	0.91	0.09
	PLS	0.82	0.30	0.86	0.18	0.82	0.22	0.85	0.19
	SVM	0.84	0.20	1.12	0.14	0.84	0.25	1.07	0.18
Leaf retention	GLMSS	0.77	0.29	0.69	0.27	0.84	0.18	0.92	0.03
	KNN	0.80	0.15	0.74	0.21	0.83	0.09	0.80	0.13
	PLS	0.79	0.13	0.69	0.27	0.76	0.05	0.77	0.11
	SVM	0.95	0.13	0.71	0.21	0.86	0.14	0.94	0.14
Above-ground biomass yield	GLMSS	7.66	0.15	7.49	0.12	7.26	0.13	6.64	0.11
	KNN	7.29	0.11	7.37	0.09	7.30	0.06	6.83	0.11
	PLS	6.94	0.08	6.96	0.17	6.92	0.17	6.83	0.12
	SVM	8.70	0.09	8.47	0.07	8.87	0.11	7.98	0.22
Fresh root yield	GLMSS	7.33	0.08	7.12	0.11	7.58	0.07	7.39	0.16
	KNN	7.29	0.09	7.18	0.17	7.26	0.06	7.18	0.06
	PLS	6.83	0.11	6.93	0.08	6.72	0.12	6.82	0.15
	SVM	8.43	0.14	8.23	0.04	8.36	0.15	8.41	0.08
Dry matter content in roots	GLMSS	1.81	0.24	1.81	0.29	1.98	0.24	2.00	0.25
	KNN	1.84	0.20	1.83	0.25	1.94	0.18	1.86	0.24
	PLS	1.78	0.34	1.76	0.27	1.90	0.16	2.04	0.10
	SVM	1.78	0.25	1.77	0.27	5.25	0.10	3.11	0.07
Leaf spot resistance	GLMSS	1.04	0.10	0.70	0.12	0.76	0.12	0.68	0.09
	KNN	0.70	0.11	0.67	0.16	0.71	0.15	0.67	0.11
	PLS	0.67	0.05	0.65	0.20	0.67	0.09	0.67	0.05
	SVM	0.74	0.14	0.92	0.08	0.85	0.13	0.77	0.09

^1^ Generalized Linear Model with Stepwise Feature Selection (GLMSS), Partial Least Squares (PLS), Support Vector Machine (SVM), K-Nearest Neighbor (KNN). RMSE: root mean square error, R^2^: coefficient of determination.

**Table 4 plants-14-00032-t004:** Vegetation indices based on the RGB and multispectral cameras as described in the table.

Description	Indices ^1^	Formula ^2^	Related Traits	Reference
Blue Green Pigment Index	BGI ^rgb, M^	$\frac{B}{G}$	Chlorophyll and leaf area index	[66]
Green Leaf Index	GLI ^rgb, M^	$\frac{\left(2G-R-B\right)}{2G+R+B}$	Chlorophyll	[67]
Normalized Green–Red Difference Index	NGRD ^rgb, M^	$\frac{\left(G-R\right)}{\left(G+R\right)}$	Chlorophyll, biomass, water content	[68]
Visible Atmospherically Resistant Index	VARI ^rgb, M^	$\frac{\left(G-R\right)}{\left(G+R+B\right)}$	Canopy, biomass, chlorophyll	[69]
Plant Senescence Reflectance Index	PSRI ^rgb, M^	$\frac{\left(R-G\right)}{\left(RE\right)}$	Chlorophyll, nitrogen, and maturation	[70]
Spectral Slope Saturation Index	SI ^rgb, M^	$\frac{\left(R-B\right)}{\left(R+B\right)}$	Saturation index	[71]
Soil Color Index	SCI ^rgb, M^	$\frac{\left(R-G\right)}{\left(R+G\right)}$	Soil color	[72]
Primary Colors Hue Index	HI ^rgb, M^	$\frac{\left(2^{*\;}R-G-B\right)}{\left(G-B\right)}$	Hue index	[71]
Hue Index	HUE ^rgb, M^	$arctan\left(\frac{2R-G-B}{30.5}(G-B)\right)$	General Hue index	[71]
Brightness Index	BI ^rgb, M^	$\frac{\left(R^{2}+G^{2}+B^{2}\right)}{3}$	Vegetation cover	[73]
Chlorophyll Index—green	CIG ^M^	$\frac{\left(NIR\right)}{\left(G\right)}-1$	Chlorophyll content	[74]
Normalized Difference Red Edge Index	NDRE ^rgb, M^	$\frac{\left(NIR-G\right)}{\left(NIR+G\right)}$	Chlorophyll content	[75]
Red-edge Chlorophyll Index	CIRE ^rgb, M^	$\frac{NIR}{RE}-1$ NIR−RE	Leaf chlorophyll content	[74]
Difference Vegetation Index	DVI ^rgb, M^	NIR−R	Nitrogen and chlorophyll	[75]
Normalized Difference Vegetation Index	NDVI ^rgb, M^	$\frac{\left(NIR-R\right)}{\left(NIR+R\right)}$	Chlorophyll, leaf area, biomass, and yield	[18]
Green Normalized Difference Vegetation Index	GNDVI ^rgb, M^	$\frac{\left(NIR-G\right)}{\left(NIR+G\right)}$	Chlorophyll, leaf area, nitrogen, and proteins	[76]
Ratio Vegetation Index	RV ^M^	$\frac{NIR}{R}$	Biomass, water, and nitrogen	[77]
Chlorophyll Vegetation Index	CVI ^M^	$\frac{\left({NIR}^{*}R\right)}{\left(G^{2}\right)}$	Chlorophyll	[78]

^1^ rgb: red–geen–blue; M: multispectral; ^2^ R: red; G: green; B: blue; NIR: near-infrared; RE: red edge.

## Data Availability

The original contributions presented in the study are included in the article and the Appendix A; further inquiries can be directed to the corresponding author.

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
