# Peer review of "High-Throughput Phenotyping for Agronomic Traits in Cassava Using Aerial Imaging"

_plants, 2024, doi:10.3390/plants14010032_

Round 1
Reviewer 1 Report
Comments and Suggestions for Authors
Dear Authors,
Thank you for your submission to Plants. After a thorough review, I believe that your study on the use of UAV-based phenotyping for cassava is a valuable contribution to the field of precision agriculture and plant breeding. However, there are several areas that require improvement before the manuscript can be considered for publication. Please address the following comments:
-
Clarification of Novelty:
- While your study is well-designed, the novelty and contribution relative to existing studies need to be more explicitly highlighted in the introduction. Please elaborate on how your work differentiates from previous research in UAV-based phenotyping, particularly in terms of its applications to cassava.
-
Reduction of Figures in the Main Text:
- The manuscript currently includes 12 figures, which makes it dense and challenging to follow. To enhance readability, I recommend moving some of the figures (e.g., Figures 6-7 and Figures 8-13) to the supplementary materials. This will allow you to maintain the focus on the most critical findings in the main text while still providing detailed data for interested readers.
-
Explanation of Model Performance:
- The results section presents various models (GLMSS, KNN, PLS, SVM) used for trait prediction. However, there is limited discussion on why certain models perform better at specific flight heights. Please provide a more detailed analysis of the performance differences among models and discuss possible reasons for the observed variations.
-
Heritability Estimates and Implications:
- Figures 4 and 5 show the broad-sense heritability estimates for different vegetation indices, which are crucial for understanding the genetic stability of traits. It would be beneficial to expand the discussion on how these estimates can influence breeding strategies and decisions for cassava.
-
Technical Details and Methodology:
- Some sections of the methodology, such as the process of radiometric calibration and the use of ground control points (GCPs), require further clarification to ensure reproducibility. Please provide more details about these procedures.
By addressing these comments, you can enhance the quality and impact of your manuscript. Once you have made the necessary revisions, please resubmit your manuscript for further consideration.
Thank you for your efforts, and I look forward to your revised submission.
Best regards,
Comments on the Quality of English Language-
Language and Clarity:
- While the manuscript is generally well-written, there are some sections where the language could be simplified for better clarity. Please review the text for any grammatical errors and improve readability.
Author Response
Thank you for your submission to Plants. After a thorough review, I believe that your study on the use of UAV-based phenotyping for cassava is a valuable contribution to the field of precision agriculture and plant breeding. However, there are several areas that require improvement before the manuscript can be considered for publication. Please address the following comments:
- Clarification of Novelty:
- While your study is well-designed, the novelty and contribution relative to existing studies need to be more explicitly highlighted in the introduction. Please elaborate on how your work differentiates from previous research in UAV-based phenotyping, particularly in terms of its applications to cassava.
Response: Thank you for your valuable contributions to the manuscript. We have highlighted the revisions in red in the final two paragraphs of the introduction.
- Reduction of Figures in the Main Text:
- The manuscript currently includes 12 figures, which makes it dense and challenging to follow. To enhance readability, I recommend moving some of the figures (e.g., Figures 6-7 and Figures 8-13) to the supplementary materials. This will allow you to maintain the focus on the most critical findings in the main text while still providing detailed data for interested readers.
Response: Ok. Figures 6–7 and 8–13 have been moved to the supplementary material.
- Explanation of Model Performance:
- The results section presents various models (GLMSS, KNN, PLS, SVM) used for trait prediction. However, there is limited discussion on why certain models perform better at specific flight heights. Please provide a more detailed analysis of the performance differences among models and discuss possible reasons for the observed variations.
Response: We have broadened the discussion on model performance to include key aspects such as spatial resolution and field of view, aligning with insights from recent literature. This expanded analysis has been incorporated into the fourth, fifth, and sixth paragraphs of the section titled “3.5. Performance of vegetation index-based models for predicting agronomic traits in cassava”.
- Heritability Estimates and Implications:
- Figures 4 and 5 show the broad-sense heritability estimates for different vegetation indices, which are crucial for understanding the genetic stability of traits. It would be beneficial to expand the discussion on how these estimates can influence breeding strategies and decisions for cassava.
Response: Ok. We have included a more comprehensive discussion on the implications of heritability magnitudes for the breeding program. These modifications can be found in the second through fourth paragraphs of the section “3.4. Heritability of vegetation indices and agronomic traits”.
- Technical Details and Methodology:
- Some sections of the methodology, such as the process of radiometric calibration and the use of ground control points (GCPs), require further clarification to ensure reproducibility. Please provide more details about these procedures.
Response: Details on the calibration process and the use of control points have been added to the Materials and Methods section.
Comments on the Quality of English Language
- Language and Clarity:
- While the manuscript is generally well-written, there are some sections where the language could be simplified for better clarity. Please review the text for any grammatical errors and improve readability.
Response: The text has been reviewed by an expert. Thank you for your suggestions.
Reviewer 2 Report
Comments and Suggestions for Authors
The whole structure was disordered, the reviewer cannot review smoothly.
1 the section titles were disordered. They were should be: introduction, materials and methods, results, discussion, conclusion.
2 the format of citing references was wrong. [1] not (1).
3 the second section should be ‘materials and methods.’ The plant variety and planting method and which stages were researched should be specified. The agronomic experiment needs great improving. In addition, data acquisition method, device and height should be specified in methods section.
Please re-arrange the structure and resubmit again.
Comments on the Quality of English LanguageModerate editing of English language required.
Author Response
The whole structure was disordered, the reviewer cannot review smoothly.
1 the section titles were disordered. They were should be: introduction, materials and methods, results, discussion, conclusion.
Response: The manuscript structure adheres to the guidelines of the journal Plants. As per the journal’s requirements, the Materials and Methods section is placed at the end of the manuscript.
2 the format of citing references was wrong. [1] not (1).
Response: The references have been reviewed and corrected according to the journal's guidelines.
3 the second section should be ‘materials and methods.’ The plant variety and planting method and which stages were researched should be specified. The agronomic experiment needs great improving. In addition, data acquisition method, device and height should be specified in methods section.
Response: The necessary changes have been made throughout the section "Materials and Methods"
Comments on the Quality of English Language
Moderate editing of English language required.
Response: The text has been reviewed by an expert. Thank you for your suggestions.
Reviewer 3 Report
Comments and Suggestions for Authors
The authors have done a lot of works trying to predict plant traits using UAV-based RGB and spectral imagery. The results can provide valuable insights into UAV flight height, selection of vegetation indices and predicting models for agronomic practices. However, I still have some concerns about the manuscript that should be addressed before it can proceed to the next step:
1. We usually use ‘unmanned aerial vehicles’ instead of ‘remotely piloted aircraft’ to name the remote sensing platform. These two terms were mix-used in the manuscript. Please unify.
2. The main text is too lengthy and should be condensed to present real useful information to readers. At first, the introduction can be improved to focus on the progress of the three objectives raised in this study, rather than presenting indirectly-related information. For example, the first three paragraphs can be combined and condensed. While research on the impact of different flight heights should be presented. Lines 80-93 should be further summarized.
3. Even though the first objective of this manuscript is to evaluate the effects of different flight heights, and since each result includes the analysis of different heights, I cannot see any usefulness of the results presented in Section 2.2, even including Figures 4-5. The effects of different heights on variation in VIs could be due to view geometry, canopy heterogeneity, etc., which is obviously not the topic of this study. And these results do not support other analyses. So, to focus on how different heights affect the prediction of plant traits, I suggest these parts either be moved to the appendix or removed.
4. Similarly, there are some inconsequential results displayed in figures 8-13. It is not necessary to publish every detail of the analysis; instead, the authors should select and present, but better to present the most critical findings that bolster the conclusions. I suggest re-organizing this part and selecting the results to ensure that only the most relevant results are highlighted.
5. Discussion should be revised to reflect the changes made in response to the above points. It should provide a more focused analysis that directly relates to the study's objectives and findings.
Comments on the Quality of English Language
Moderate editing of English language required.
Author Response
I still have some concerns about the manuscript that should be addressed before it can proceed to the next step:
- We usually use ‘unmanned aerial vehicles’ instead of ‘remotely piloted aircraft’ to name the remote sensing platform. These two terms were mix-used in the manuscript. Please unify.
Response: Ok. We have replaced "remotely piloted aircraft" with "unmanned aerial vehicles" throughout the entire text.
- The main text is too lengthy and should be condensed to present real useful information to readers. At first, the introduction can be improved to focus on the progress of the three objectives raised in this study, rather than presenting indirectly-related information. For example, the first three paragraphs can be combined and condensed. While research on the impact of different flight heights should be presented. Lines 80-93 should be further summarized.
Response: The introduction has been reorganized and revised according to the suggestions.
- Even though the first objective of this manuscript is to evaluate the effects of different flight heights, and since each result includes the analysis of different heights, I cannot see any usefulness of the results presented in Section 2.2, even including Figures 4-5. The effects of different heights on variation in VIs could be due to view geometry, canopy heterogeneity, etc., which is obviously not the topic of this study. And these results do not support other analyses. So, to focus on how different heights affect the prediction of plant traits, I suggest these parts either be moved to the appendix or removed.
Response: Section 2.2 has been removed from the text.
- Similarly, there are some inconsequential results displayed in figures 8-13. It is not necessary to publish every detail of the analysis; instead, the authors should select and present, but better to present the most critical findings that bolster the conclusions. I suggest re-organizing this part and selecting the results to ensure that only the most relevant results are highlighted.
Response: Corrected. Figures 8 to 13 have been moved to the supplementary material section.
- Discussion should be revised to reflect the changes made in response to the above points. It should provide a more focused analysis that directly relates to the study's objectives and findings.
Response: Corrected. We have made the revisions to the discussion in response to the reviewers' comments.
Round 2
Reviewer 1 Report
Comments and Suggestions for Authors
Dear Authors,
Thank you for your comprehensive revisions, which have significantly improved the manuscript. The study on UAV-based high-throughput phenotyping in cassava provides valuable insights for precision agriculture and plant breeding. However, there are a few minor issues that need to be addressed to further enhance the quality and clarity of the manuscript before it can be accepted for publication.
Methodological Details:
The inclusion of radiometric calibration and ground control point (GCP) procedures in the Materials and Methods section is appreciated. However, to ensure full reproducibility, please consider providing additional details on these processes, such as specific calibration settings, equipment used, or step-by-step descriptions. This added clarity will strengthen the rigor and reproducibility of your study.
Figure and Table Presentation:
The reorganization of figures to the supplementary material has improved the manuscript's readability. However, Figures 4 and 5, which discuss broad-sense heritability estimates, could still benefit from more explicit labeling and clearer figure legends. Ensuring that the figures are self-explanatory will greatly aid in the reader’s understanding of the presented data.
Additionally, please review all figure captions for consistency and consider adding brief interpretations where necessary to guide the reader.
Discussion on Model Performance:
The expanded discussion on model performance is well-received. Nevertheless, further elaboration on why certain models (e.g., GLMSS, KNN) perform differently across flight heights would be beneficial. Specifically, discussing how factors such as spatial resolution, field of view, and environmental variability influence model outcomes will add depth to the analysis and contextualize the predictive capabilities of each approach.
Comments on the Quality of English LanguageLanguage and Readability:
Although the manuscript has undergone language editing, there are still sections where the clarity and readability could be further improved. Simplifying complex sentences and refining certain technical descriptions will help make the manuscript more accessible to a broader audience. For instance, some of the results and methodological explanations could benefit from a more concise and direct presentation.
Author Response
Thank you for your comprehensive revisions, which have significantly improved the manuscript. The study on UAV-based high-throughput phenotyping in cassava provides valuable insights for precision agriculture and plant breeding. However, there are a few minor issues that need to be addressed to further enhance the quality and clarity of the manuscript before it can be accepted for publication.
Methodological Details:
The inclusion of radiometric calibration and ground control point (GCP) procedures in the Materials and Methods section is appreciated. However, to ensure full reproducibility, please consider providing additional details on these processes, such as specific calibration settings, equipment used, or step-by-step descriptions. This added clarity will strengthen the rigor and reproducibility of your study.
Response: Additional details on radiometric calibration and equipment have been included in the methodology, at the end of sections 5.3 and 5.4.
Figure and Table Presentation:
The reorganization of figures to the supplementary material has improved the manuscript's readability. However, Figures 4 and 5, which discuss broad-sense heritability estimates, could still benefit from more explicit labeling and clearer figure legends. Ensuring that the figures are self-explanatory will greatly aid in the reader’s understanding of the presented data.
Additionally, please review all figure captions for consistency and consider adding brief interpretations where necessary to guide the reader.
Response: The figure captions have been revised, and corrections were made in accordance with the recommendations.
Discussion on Model Performance:
The expanded discussion on model performance is well-received. Nevertheless, further elaboration on why certain models (e.g., GLMSS, KNN) perform differently across flight heights would be beneficial. Specifically, discussing how factors such as spatial resolution, field of view, and environmental variability influence model outcomes will add depth to the analysis and contextualize the predictive capabilities of each approach.
Response: The discussion was enhanced by linking the effects of spatial resolution, field of view, and environmental variability with flight altitude to the performance of the models.
Comments on the Quality of English Language
Language and Readability:
Although the manuscript has undergone language editing, there are still sections where the clarity and readability could be further improved. Simplifying complex sentences and refining certain technical descriptions will help make the manuscript more accessible to a broader audience. For instance, some of the results and methodological explanations could benefit from a more concise and direct presentation.
Response: The results and methodology were adjusted following the recommendations of the three reviewers.
Reviewer 2 Report
Comments and Suggestions for Authors
Comments on the Quality of English Language
Moderate editing of English language required.
Author Response
Moderate editing of English language required.
Response: We made the necessary adjustments to make the article more accessible to a broader audience.
Reviewer 3 Report
Comments and Suggestions for Authors
The manuscript has been improved. I have no more comments about it except for some language errors that should be corrected.
Comments on the Quality of English LanguageMinor editing of English language still required.
Author Response
The manuscript has been improved. I have no more comments about it except for some language errors that should be corrected.
Comments on the Quality of English Language
Minor editing of English language still required.
Response: We made the necessary adjustments to make the article more accessible to a broader audience.